# Woodbury Transformations for Deep Generative Flows

**You Lu**
Department of Computer Science
Virginia Tech
Blacksburg, VA
you.lu@vt.edu

**Bert Huang**
Department of Computer Science
Tufts University
Medford, MA
bert@cs.tufts.edu

## Abstract

Normalizing flows are deep generative models that allow efficient likelihood calculation and sampling. The core requirement for this advantage is that they are constructed using functions that can be efficiently inverted and for which the determinant of the function's Jacobian can be efficiently computed. Researchers have introduced various such flow operations, but few of these allow rich interactions among variables without incurring significant computational costs. In this paper, we introduce *Woodbury transformations*, which achieve efficient invertibility via the Woodbury matrix identity and efficient determinant calculation via Sylvester's determinant identity. In contrast with other operations used in state-of-the-art normalizing flows, Woodbury transformations enable (1) high-dimensional interactions, (2) efficient sampling, and (3) efficient likelihood evaluation. Other similar operations, such as 1x1 convolutions, emerging convolutions, or periodic convolutions allow at most two of these three advantages. In our experiments on multiple image datasets, we find that Woodbury transformations allow learning of higher-likelihood models than other flow architectures while still enjoying their efficiency advantages.

## 1 Introduction

Deep generative models are powerful tools for modeling complex distributions and have been applied to many tasks such as synthetic data generation [26, 37], domain adaption [38], and structured prediction [32]. Examples of these models include autoregressive models [13, 27], variational autoencoders [20, 30], generative adversarial networks [11], and normalizing flows [6, 7, 21, 29]. Normalizing flows are special because of two advantages: They allow efficient and exact computation of log-likelihood and sampling.

Flow-based models are composed of a series of invertible functions, which are specifically designed so that their inverse and determinant of the Jacobian are easy to compute. However, to preserve this computational efficiency, these functions usually cannot sufficiently encode dependencies among dimensions of a variable. For example, affine coupling layers [6] split a variable to two parts and require the second part to only depend on the first. But they ignore the dependencies among dimensions in the second part.

To address this problem, Dinh et al. [6, 7] introduced a fixed permutation operation that reverses the ordering of the channels of pixel variables. Kingma and Dhariwal [21] introduced a $1 \times 1$ convolution, which are a generalized permutation layer, that uses a weight matrix to model the interactions among dimensions along the channel axis. Their experiments demonstrate the importance of capturing dependencies among dimensions. Relatedly, Hoogeboom et al. [15] proposed emerging convolution

operations, and Hoogeboom et al. [15] and Finz et al. [9] proposed periodic convolution. These two convolution layers have $d \times d$ kernels that can model dependencies along the spatial axes in addition to the channel axis. However, the increase in representational power comes at a cost: These convolution operations do not scale well to high-dimensional variables. The emerging convolution is a combination of two autoregressive convolutions [10, 22], whose inverse is not parallelizable. To compute the inverse or determinant of the Jacobian, the periodic convolution requires transforming the input and the convolution kernel to Fourier space. This transformation is computationally costly.

In this paper, we develop *Woodbury transformations* for generative flows. Our method is also a generalized permutation layer and uses spatial and channel transformations to model dependencies among dimensions along spatial and channel axes. We use the Woodbury matrix identity [36] and Sylvester's determinant identity [34] to compute the inverse and Jacobian determinant, respectively, so that both the training and sampling time complexities are linear to the input variable's size. We also develop a memory-efficient variant of the Woodbury transformation, which has the same advantage as the full transformation but uses significantly reduced memory when the variable is high-dimensional. In our experiments, we found that Woodbury transformations enable model quality comparable to many state-of-the-art flow architectures while maintaining significant efficiency advantages.

## 2 Deep Generative Flows

In this section, we briefly introduce the deep generative flows. More background knowledge can be found in the appendix.

A normalizing flow [29] is composed of a series of invertible functions $\mathbf{f} = \mathbf{f}_1 \circ \mathbf{f}_2 \circ ... \circ \mathbf{f}_K$, which transform $\mathbf{x}$ to a latent code $\mathbf{z}$ drawn from a simple distribution. Therefore, with the *change of variables* formula, we can rewrite the log-likelihood $\log p_\theta(\mathbf{x})$ to be

$$\log p_\theta(\mathbf{x}) = \log p_Z(\mathbf{z}) + \sum_{i=1}^{K} \log \left| \det \left( \frac{\partial \mathbf{f}_i}{\partial \mathbf{r}_{i-1}} \right) \right|, \tag{1}$$

where $\mathbf{r}_i = \mathbf{f}_i(\mathbf{r}_{i-1})$, $\mathbf{r}_0 = \mathbf{x}$, and $\mathbf{r}_K = \mathbf{z}$.

Flow-based generative models [6, 7, 21] are developed on the theory of normalizing flows. Each transformation function used in the models is a specifically designed neural network that has a tractable Jacobian determinant and inverse. We can sample from a trained flow $\mathbf{f}$ by computing $\mathbf{z} \sim p_Z(\mathbf{z})$, $\quad \mathbf{x} = \mathbf{f}^{-1}(\mathbf{z})$.

There have been many operations, i.e., layers, proposed in recent years for generative flows. In this section, we discuss some commonly used ones, and more related works will be discussed in Section 4.

**Actnorm layers** [21] perform per-channel affine transformations of the activations using scale and bias parameters to improve training stability and performance. The actnorm is formally expressed as $\mathbf{y}_{:,i,j} = \mathbf{s} \odot \mathbf{x}_{:,i,j} + \mathbf{b}$, where both the input $\mathbf{x}$ and the output $\mathbf{y}$ are $c \times h \times w$ tensors, $c$ is the channel dimension, and $h \times w$ are spatial dimensions. The parameters $\mathbf{s}$ and $\mathbf{b}$ are $c \times 1$ vectors.

**Affine coupling layers** [6, 7] split the input $\mathbf{x}$ into two parts, $\mathbf{x}_a, \mathbf{x}_b$. And then fix $\mathbf{x}_a$ and force $\mathbf{x}_b$ to only relate to $\mathbf{x}_a$, so that the Jacobian is a triangular matrix. Formally, we compute

$$\mathbf{x}_a, \mathbf{x}_b = \text{split}(\mathbf{x}), \qquad \mathbf{y}_a = \mathbf{x}_a,$$
$$\mathbf{y}_b = \mathbf{s}(\mathbf{x}_a) \odot \mathbf{x}_b + \mathbf{b}(\mathbf{x}_a), \quad \mathbf{y} = \text{concat}(\mathbf{y}_a, \mathbf{y}_b),$$

where $\mathbf{s}$ and $\mathbf{b}$ are two neural networks with $\mathbf{x}_a$ as input. The split and the concat split and concatenate the variables along the channel axis. Usually, $s$ is restricted to be positive. An additive coupling layer is a special case when $\mathbf{s} = \mathbf{1}$.

Actnorm layers only rescale the dimensions of $\mathbf{x}$, and affine coupling layers only relate $\mathbf{x}_b$ to $\mathbf{x}_a$ but omit dependencies among different dimensions of $\mathbf{x}_b$. Thus, we need other layers to capture local dependencies among dimensions.

**Invertible convolutional layers** [9, 15, 21] are generalized permutation layers that can capture correlations among dimensions. The $1 \times 1$ convolution [21] is $\mathbf{y}_{:,i,j} = \mathbf{M}\mathbf{x}_{:,i,j}$, where $\mathbf{M}$ is a $c \times c$ matrix. The Jacobian of a $1 \times 1$ convolution is a block diagonal matrix, so that its log-determinant is $hw \log |\det(\mathbf{M})|$. Note that the $1 \times 1$ convolution only operates along the channel axis and ignores the dependencies along the spatial axes.

Emerging convolutions [15] combine two autoregressive convolutions [10, 22]. Each autoregressive convolution masks out some weights to force an autoregressive structure, so that the Jacobian is a triangular matrix and computing its determinant is efficient. One problem of emerging convolution is the computation of inverse is non-parallelizable, so that is ineffecent for high-dimensional variables.

Periodic convolutions [9, 15] transform the input and kernel to the Fourier domain using discrete Fourier transformations, so the convolution function is an element-wise matrix product with a block-diagonal Jacobian. The computational cost of periodic convolutions is $\mathcal{O}(chw \log(hw) + c^3 hw)$. Thus, when the input is high-dimensional, both training and sampling are expensive.

**Multi-scale architectures** [7] compose flow layers to generate rich models, using *split layers* to factor out variables and *squeeze layers* to shuffle dimensions, resulting in an architecture with $K$ flow steps and $L$ levels. See Fig. 1.

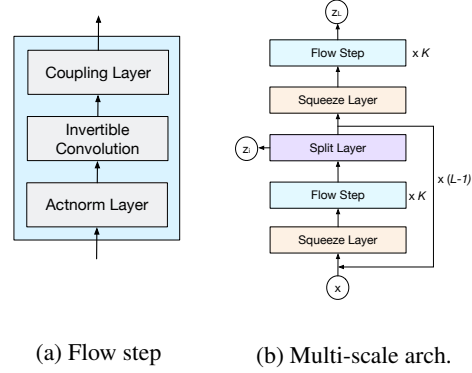

(a) Flow step      (b) Multi-scale arch.

Figure 1: Overview of architecture of generative flows. We can design the flow step by selecting a suitable convolutional layer and a coupling layer based on the task. Glow [21] uses $1 \times 1$ convolutions and affine coupling.

# 3 Woodbury Transformations

In this section, we introduce Woodbury transformations as an efficient means to model high-dimensional correlations.

## 3.1 Channel and Spatial Transformations

Suppose we reshape the input $\mathbf{x}$ to be a $c \times n$ matrix, where $n = hw$. Then the $1 \times 1$ convolution can be reinterpreted as a matrix transformation

$$\mathbf{y} = \mathbf{W}^{(c)} \mathbf{x}, \tag{2}$$

where $\mathbf{y}$ is also a $c \times n$ matrix, and $\mathbf{W}^{(c)}$ is a $c \times c$ matrix. For consistency, we will call this a channel transformation. For each column $\mathbf{x}_{:,i}$, the correlations among channels are modeled by $\mathbf{W}^{(c)}$. However, the correlation between any two rows $\mathbf{x}_{:,i}$ and $\mathbf{x}_{:,j}$ is not captured. Inspired by Eq. 2, we use a spatial transformation to model interactions among dimensions along the spatial axis

$$\mathbf{y} = \mathbf{x} \mathbf{W}^{(s)}, \tag{3}$$

where $\mathbf{W}^{(s)}$ is an $n \times n$ matrix that models the correlations of each row $\mathbf{x}_{i,:}$. Combining Equation 2 and Equation 3, we have

$$\mathbf{x}_c = \mathbf{W}^{(c)} \mathbf{x}, \qquad \mathbf{y} = \mathbf{x}_c \mathbf{W}^{(s)}. \tag{4}$$

For each dimension of output $\mathbf{y}_{i,j}$, we have $\mathbf{y}_{i,j} = \sum_{v=1}^{c} \left( \sum_{u=1}^{n} \mathbf{W}_{i,u}^{(c)} \cdot \mathbf{x}_{u,v} \right) \cdot \mathbf{W}_{v,j}^{(s)}$.

Therefore, the spatial and channel transformations together can model the correlation between any pair of dimensions. However, in this preliminary form, directly using Eq. 4 is inefficient for large $c$ or $n$. First, we would have to store two large matrices $\mathbf{W}^c$ and $\mathbf{W}^s$, so the space cost is $\mathcal{O}(c^2 + n^2)$. Second, the computational cost of Eq. 4 is $\mathcal{O}(c^2 n + n^2 c)$—quadratic in the input size. Third, the computational cost of the Jacobian determinant is $\mathcal{O}(c^3 + n^3)$, which is far too expensive in practice.

## 3.2 Woodbury Transformations

We solve the three scalability problems by using a low-rank factorization. Specifically, we define

$$\mathbf{W}^{(c)} = \mathbf{I}^{(c)} + \mathbf{U}^{(c)} \mathbf{V}^{(c)}, \qquad \mathbf{W}^{(s)} = \mathbf{I}^{(s)} + \mathbf{U}^{(s)} \mathbf{V}^{(s)},$$

where $\mathbf{I}^{(c)}$ and $\mathbf{I}^{(s)}$ are $c$- and $n$-dimensional identity matrices, respectively. The matrices $\mathbf{U}^c$, $\mathbf{V}^c$, $\mathbf{U}^s$, and $\mathbf{V}^s$ are of size $c \times d_c$, $d_c \times c$, $n \times d_s$, and $d_c \times n$, respectively, where $d_c$ and $d_s$ are constant latent dimensions of these four matrices. Therefore, we can rewrite Equation 4 as

$$\mathbf{x}_c = (\mathbf{I}^{(c)} + \mathbf{U}^{(c)}\mathbf{V}^{(c)})\mathbf{x}, \qquad \mathbf{y} = \mathbf{x}_c(\mathbf{I}^{(s)} + \mathbf{U}^{(s)}\mathbf{V}^{(s)}). \tag{5}$$

We call Eq. 5 the Woodbury transformation because the Woodbury matrix identity [36] and Sylvester's determinant identity [34] allow efficient computation of its inverse and Jacobian determinant.

**Woodbury matrix identity.**[1] Let $\mathbf{I}^{(n)}$ and $\mathbf{I}^{(k)}$ be $n$- and $k$-dimensional identity matrices, respectively. Let $\mathbf{U}$ and $\mathbf{V}$ be $n \times k$ and $k \times n$ matrices, respectively. If $\mathbf{I}^{(k)} + \mathbf{VU}$ is invertible, then $(\mathbf{I}^{(n)} + \mathbf{UV})^{-1} = \mathbf{I}^{(n)} - \mathbf{U}(\mathbf{I}^k + \mathbf{VU})^{-1}\mathbf{V}$.

**Sylvester's determinant identity.** Let $\mathbf{I}^{(n)}$ and $\mathbf{I}^{(k)}$ be $n$- and $k$-dimensional identity matrices, respectively. Let $\mathbf{U}$ and $\mathbf{V}$ be $n \times k$ and $k \times n$ matrices, respectively. Then, $\det(\mathbf{I}^{(n)} + \mathbf{UV}) = \det(\mathbf{I}^{(k)} + \mathbf{VU})$.

Based on these two identities, we can efficiently compute the inverse and Jacobian determinant

$$\begin{aligned}
\mathbf{x}_c &= \mathbf{y}(\mathbf{I}^{(s)} - \mathbf{U}^{(s)}(\mathbf{I}^{(d_s)} + \mathbf{V}^{(s)}\mathbf{U}^{(s)})^{-1}\mathbf{V}^{(s)}), \\
\mathbf{x} &= (\mathbf{I}^{(c)} - \mathbf{U}^{(c)}(\mathbf{I}^{(d_c)} + \mathbf{V}^{(c)}\mathbf{U}^{(c)})^{-1}\mathbf{V}^{(c)})\mathbf{x}_c,
\end{aligned} \tag{6}$$

and

$$\log\left|\det\left(\frac{\partial \mathbf{y}}{\partial \mathbf{x}}\right)\right| = n \log\left|\det(\mathbf{I}^{(d_c)} + \mathbf{V}^{(c)}\mathbf{U}^{(c)})\right| + c \log\left|\det(\mathbf{I}^{(d_s)} + \mathbf{V}^{(s)}\mathbf{U}^{(s)})\right|, \tag{7}$$

where $\mathbf{I}^{(d_c)}$ and $\mathbf{I}^{(d_s)}$ are $d_c$- and $d_s$-dimensional identity matrices, respectively.

A Woodbury transformation is also a generalized permutation layer. We can directly replace an invertible convolution in Figure 1a with a Woodbury transformation. In contrast with $1 \times 1$ convolutions, Woodbury transformations are able to model correlations along both channel and spatial axes. We illustrate this in Figure 2. To implement Woodbury transformations, we need to store four weight matrices, i.e., $\mathbf{U}^{(c)}, \mathbf{U}^{(s)}, \mathbf{V}^{(c)}$, and $\mathbf{V}^{(s)}$. To simplify our analysis, let $d_c \leq d$ and $d_s \leq d$, where $d$ is a constant. This setting is also consistent with our experiments. The size of $\mathbf{U}^{(c)}$ and $\mathbf{V}^{(c)}$ is $\mathcal{O}(dc)$, and the size of $\mathbf{U}^{(c)}$ and $\mathbf{V}^{(c)}$ is $\mathcal{O}(dn)$. The space complexity is $\mathcal{O}(d(c + n))$.

For training and likelihood computation, the main computational bottleneck is computing $\mathbf{y}$ and the Jacobian determinant. To compute $\mathbf{y}$ with Equation 4, we need to first compute the channel transformation and then compute the spatial transformation. The computational complexity is $\mathcal{O}(dcn)$. To compute the determinant with Equation 7, we need to first compute the matrix product of $\mathbf{V}$ and $\mathbf{U}$, and then compute the determinant. The computational complexity is $\mathcal{O}(d^2(c + n) + d^3)$.

For sampling, we need to compute the inverse transformations, i.e., Equation 6. With the Woodbury identity, we actually only need to compute the inverses of $\mathbf{I}^{(d_s)} + \mathbf{V}^{(s)}\mathbf{U}^{(s)}$ and $\mathbf{I}^{(d_c)} + \mathbf{V}^{(c)}\mathbf{U}^{(c)}$, which are computed with time complexity $\mathcal{O}(d^3)$. To implement the inverse transformations, we can compute the matrix chain multiplication, so we can avoid computing the product of two large matrices twice, yielding cost $\mathcal{O}(c^2 + n^2)$. For example, for the inverse spatial transformation, we can compute it as $\mathbf{x}_c = \mathbf{y} - ((\mathbf{y}\mathbf{U}^{(s)})(\mathbf{I}^{(d_s)} + \mathbf{V}^{(s)}\mathbf{U}^{(s)})^{-1})\mathbf{V}^{(s)}$, so that its complexity is $\mathcal{O}(d^3 + cd^2 + cnd)$. The total computational complexity of Equation 6 is $\mathcal{O}(dcn + d^2(n + c) + d^3)$.

In practice, we found that for a high-dimensional input, a relatively small $d$ is enough to obtain good performance, e.g., the input is $256 \times 256 \times 3$ images, and $d = 16$. In this situation, $nc \geq d^3$. Therefore, we can omit $d$ and approximately see the spatial complexity as $\mathcal{O}(c + n)$, and the forward or inverse transformation as $\mathcal{O}(nc)$. They are all linear to the input size.

We do not restrict $\mathbf{U}$ and $\mathbf{V}$ to force $\mathbf{W}$ to be invertible. Based on analysis by Hoogeboom et al. [15], the training maximizes the log-likelihood, which implicitly pushes $\det(\mathbf{I} + \mathbf{VU})$ away from 0. Therefore, it is not necessary to explicitly force invertibility. In our experiments, the Woodbury transformations are as robust as other invertible convolution layers.

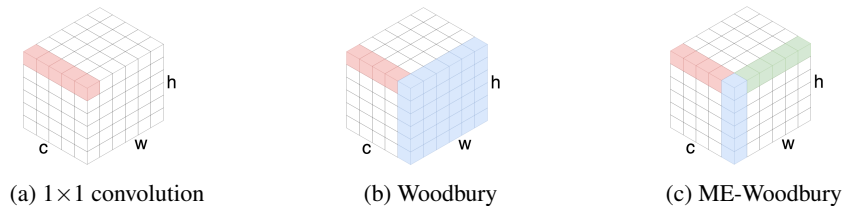

|  (a) 1×1 convolution | (b) Woodbury | (c) ME-Woodbury |

Figure 2: Visualization of three transformations. The 1×1 convolution only operates along the channel axis. The Woodbury transformation operates along both the channel and spatial axes, modeling the dependencies of one channel directly via one transformation. The ME-Woodbury transformation operates along three axes. It uses two transformations to model spatial dependencies.

## 3.3 Memory-Efficient Variant

In Eq. 4, one potential challenge arises from the sizes of $\mathbf{U}^{(s)}$ and $\mathbf{V}^{((s))}$, which are linear in $n$. The challenge is that $n$ may be large in some practical problems, e.g., high-resolution images. We develop a memory-efficient variant of Woodbury transformations, i.e., ME-Woodbury, to solve this problem. The ME version can effectively reduce space complexity from $\mathcal{O}(d(c + hw))$ to $\mathcal{O}(d(c + h + w))$.

The difference between ME-Woodbury transformations and Woodbury transformations is that the ME form cannot directly model spatial correlations. As shown in Figure 2c, it uses two transformations, for height and width, together to model the spatial correlations. Therefore, for a specific channel $k$, when two dimensions $\mathbf{x}_{k,i,j}$ and $\mathbf{x}_{k,u,v}$ are in two different heights, and widths, their interaction will be modeled indirectly. In our experiments, we found that this limitation only slightly impacts ME-Woodbury's performance. More details on ME-Woodbury transformations are in the appendix.

## 4 Related Work

Rezende and Mohamed [29] developed planar flows for variational inference $\mathbf{z}_{t+1} = \mathbf{z}_t + \mathbf{u}\delta(\mathbf{w}^T\mathbf{z}_t + b)$, where $\mathbf{z}$, $\mathbf{w}$, and $\mathbf{u}$ are $d$-dimensional vectors, $\delta()$ is an activation function, and $b$ is a scalar.

Berg et al. [3] generalized these to Sylvester flows $\mathbf{z}_{t+1} = \mathbf{z}_t + \mathbf{QR}\delta(\tilde{\mathbf{R}}\mathbf{Q}^T\mathbf{z}_t + \mathbf{r})$, where $\mathbf{R}$ and $\tilde{\mathbf{R}}$ are upper triangular matrices, $\mathbf{Q}$ is composed of a set of orthonormal vectors, and $\mathbf{r}$ is a $d$-dimensional vector. The resulting Jacobian determinant can be efficiently computed via Sylvester's identity, just as our methods do. However, Woodbury transformations have key differences from Sylvester flows. First, Berg et al. only analyze their models on vectors. The inputs to our layers are matrices, so our method operates on high-dimensional input, e.g., images. Second, though Sylvester flows are inverse functions, computing their inverse is difficult. One possible way is to apply iterative methods [2, 5, 33] to compute the inverse. But this research direction is unexplored. Our layers can be inverted efficiently with the Woodbury identity. Third, our layers do not restrict the transformation matrices to be triangular or orthogonal. In fact, Woodbury transformations can be seen as another generalized variant of planar flows on matrices, with $\delta(\mathbf{x}) = \mathbf{x}$, and whose inverse is tractable. Roughly speaking, Woodbury transformations can also be viewed as applying the planar flows sequentially to each row of the input matrix. After this work was completed and submitted, we learned that the TensorFlow software [1] also uses the Woodbury identity in their affine bijector.

Normalizing flows have also been used for variational inference, density estimation, and generative modeling. Autoregressive flows [17, 22, 24, 28] restrict each variable to depend on those that precede it in a sequence, forcing a triangular Jacobian. Non-linear coupling layers replace the affine transformation function. Specifically, spline flows [8, 25] use spline interpolation, and Flow++ [14] uses a mixture cumulative distribution function to define these functions. Flow++ also uses variational dequantization to prevent model collapse. Many works [9, 15, 18, 21] develop invertible convolutional flows to model interactions among dimensions. MintNet [33] is a flexible architecture composed of multiple masked invertible layers. I-ResNet [2, 5] uses discriminative deep network architecture as the flow. These two models require iterative methods to compute the inverse. Discrete flows [16, 35] and latent flows [39] can be applied to discrete data such as text. Continuous-time flows [4, 12] have been developed based on the theory of ordinary differential equations.

## 5 Experiments

In this section, we compare the performance of Woodbury transformations against other modern flow architectures, measuring running time, bit per-dimension ($\log_2$-likelihood), and sample quality.

**Running Time** We follow Finz et al. [9] and compare the per-sample running time of Woodbury transformations to other generalized permutations: $1\times1$ [21], emerging [15], and periodic convolutions [9, 15]. We test the training time and sampling time. In training, we compute (1) forward propagation, i.e., $\mathbf{y} = \mathbf{f}(\mathbf{x})$, of a given function $\mathbf{f}()$, (2) the Jacobian determinant, i.e., $\det\left(\left|\frac{\partial \mathbf{y}}{\partial \mathbf{x}}\right|\right)$, and (3) the gradient of parameters. For sampling, we compute the inverse of transformation $\mathbf{x} = \mathbf{f}^{-1}(\mathbf{y})$. For emerging and periodic convolutions, we use $3 \times 3$ kernels. For Woodbury transformations, we fix the latent dimension $d = 16$. For fair comparison, we implement all methods in Pytorch and run them on an Nvidia Titan V GPU. We follow Hoogeboom et al. [15] and implement the emerging convolution inverse in Cython, and we compute it on a 4 Ghz CPU (the GPU version is slower than the Cython version). We first fix the spatial size to be $64 \times 64$ and vary the channel number. We then fix the channel number to be 96 and vary the spatial size.

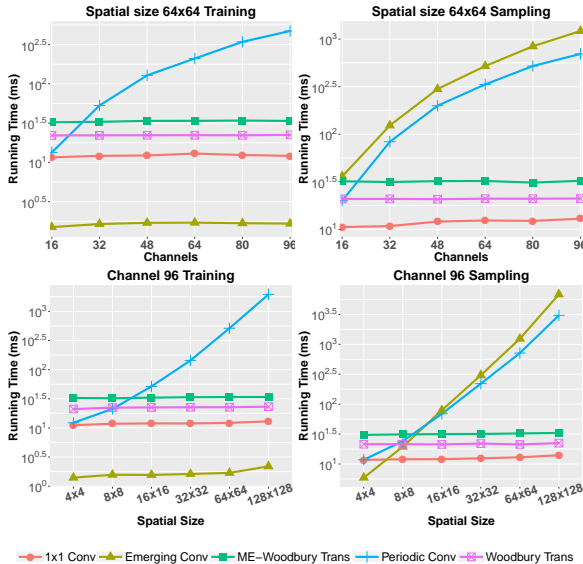

Figure 3: Running time comparison. Sampling with emerging convolutions is slow, since their inverses are not parallelizable. Periodic convolutions are costly for larger inputs. Both $1\times1$ convolutions and Woodbury transformations are efficient in training and sampling.

The results are shown in Figure 3. For training, the emerging convolution is the fastest. This is because its Jacobian is a triangular matrix, so computing its determinant is much more efficient than other methods. The Woodbury transformation and ME-Woodbury are slightly slower than the 1x1 convolution, since they contain more transformations. Emerging convolutions, Woodbury transformations, and 1x1 convolutions only slightly increase with input size, rather than increasing with $\mathcal{O}(c^3)$. This invariance to input size is likely because of how the GPU parallelizes computation. The periodic convolution is efficient only when the input size is small. When the size is large, it becomes slow, e.g., when the input size is $96 \times 64 \times 64$, it is around 30 times slower than Woodbury transformations. In our experiments, we found that the Fourier transformation requires a large amount of memory. According to Finz et al. [9], the Fourier step may be the bottleneck that impacts periodic convolution's scalability. A more efficient implementation of Fourier transformation, e.g., [18], may improve its running time.

For sampling, both $1\times1$ convolutions and Woodbury transformations are efficient. The $1\times1$ convolution is the fastest, and the Woodbury transformations are only slightly slower. Neither is sensitive to the change of input size. Emerging convolutions and periodic convolutions are much slower than Woodbury transformations, and their running time increases with the input size. When the input size is $96 \times 128 \times 128$, they are around 100 to 200 times slower than Woodbury transformations. This difference is because emerging convolutions cannot make use of parallelization, and periodic transformations require conversion to Fourier form. Based on these results, we can conclude that both emerging convolution and periodic convolution do not scale well to high-dimensional inputs. In contrast, Woodbury transformations are efficient in both training and sampling.

**Quantitative Evaluation** We compare Woodbury transformations with state-of-the-art flow models, measuring bit per-dimension (bpd). We train with the CIFAR-10 [23] and ImageNet [31] datasets. We compare with three generalized permutation methods—$1\times1$ convolution, emerging convolution, and periodic convolution—and two coupling layers—neural spline coupling [8] and MaCow [24]. We use Glow (Fig. 1, [21]) as the basic flow architecture. For each method, we replace the corresponding layer. For example, to construct a flow with Woodbury transformations, we replace the $1\times1$ convolution

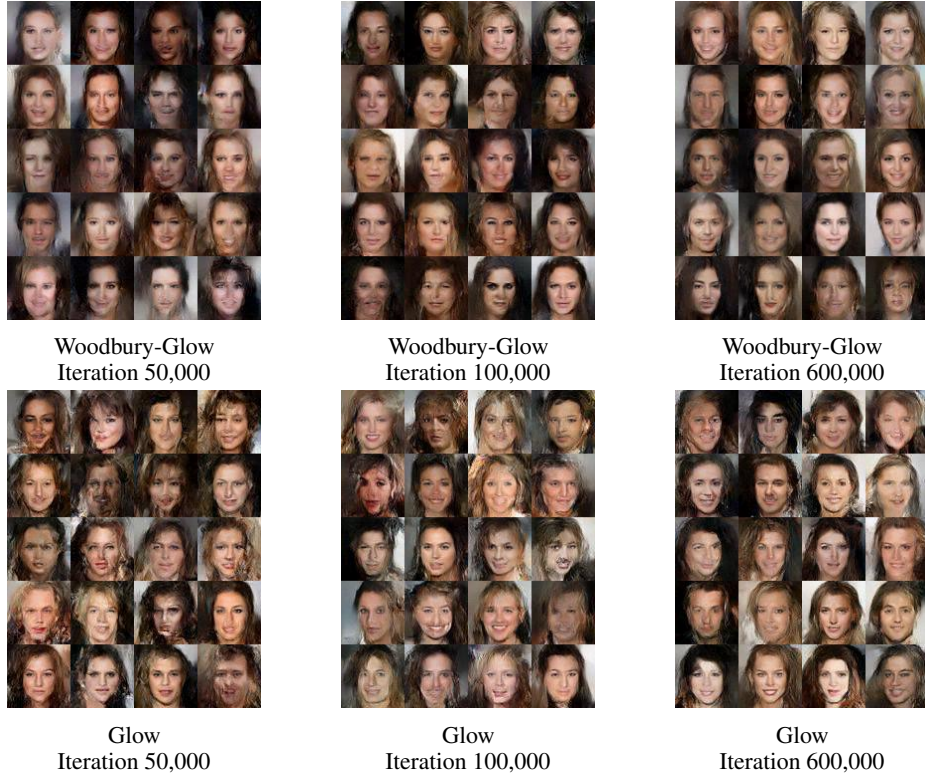

<div align="center">

Woodbury-Glow       Woodbury-Glow       Woodbury-Glow

Iteration 50,000       Iteration 100,000       Iteration 600,000

Glow       Glow       Glow

Iteration 50,000       Iteration 100,000       Iteration 600,000

</div>

Figure 4: Random samples $64 \times 64$ drawn from models trained on CelebA with temperature $0.7$.

with a Woodbury transformation, i.e., Eq. 4. For all generalized permutation methods, we use affine coupling. For each of the coupling layer baselines, we substitute it for the affine coupling. We tune the parameters of neural spline coupling and MaCow so that their sizes are close to affine coupling. We follow Hoogeboom et al. [15] and test the performance of small models. For $32 \times 32$ images, we set the number of levels to $L = 3$ and the number of steps per-level to $K = 8$. For $64 \times 64$ images, we use $L = 4$ and $K = 16$. More details are in the appendix.

<div align="center">

Table 1: Quantitative evaluation results.

</div>

| | **Quantitative measure (bpd)** | | | **Model sizes (# parameters)** | |
|---|---|---|---|---|---|
| | CIFAR-10 32x32 | ImageNet 32x32 | ImageNet 64x64 | 32x32 images | 64x64 images |
| $1\times1$ convolution | 3.51 | 4.32 | 3.94 | 11.02M | 37.04M |
| Emerging | 3.48 | 4.26 | 3.91 | 11.43M | 40.37M |
| Periodic | 3.49 | 4.28 | 3.92 | 11.21M | 38.61M |
| Neural spline | 3.50 | 4.24 | 3.95 | **10.91M** | 38.31M |
| MaCow | 3.48 | 4.34 | 4.15 | 11.43M | 37.83M |
| ME-Woodbury | 3.48 | 4.22 | 3.91 | 11.02M | **36.98M** |
| Woodbury | **3.47** | **4.20** | **3.87** | 11.10M | 37.60M |

The test-set likelihoods are listed in Table 1 left. Our scores are worse than those reported by Hoogeboom et al. [15], Kingma and Dhariwal [21] because we use smaller models and train each model on a single GPU. Based on the scores, $1\times1$ convolutions perform the worst. Emerging convolutions and periodic convolutions score better than the $1\times1$ convolutions, since they are more flexible and can model the dependencies along the spatial axes. Neural spline coupling works well on $32 \times 32$ images, but do slightly worse than $1\times1$ convolution on $64 \times 64$ images. MaCow does not work well on ImageNet. This trend demonstrates the importance of permutation layers. They can model the interactions among dimensions and shuffle them, which coupling layers cannot do. Without a good permutation layer, a better coupling layer still cannot always improve the performance. The

Woodbury transformation models perform the best, likely because they can model the interactions between the target dimension and all other dimensions, while the invertible convolutions only model the interactions between target dimension its neighbors. ME-Woodbury performs only slightly worse than the full version, showing that its restrictions provide a useful tradeoff between model quality and efficiency.

We list model sizes in Table 1 (right). Despite modeling rich interactions, Woodbury transformations are not the largest. With $32 \times 32$ images, ME-Woodbury and $1\times1$ convolution are the same size. When the image size is $64 \times 64$, ME-Woodbury is the smallest. This is because we use the multi-scale architecture, i.e., Fig. 1, to combine layers. The squeeze layer doubles the input variable's channels at each level, so larger $L$ suggests larger $c$. The space complexities of invertible convolutions are $\mathcal{O}(c^2)$, while the space complexity of ME-Woodbury is linear to $c$. When $c$ is large, the weight matrices of invertible convolutions are larger than the weight matrices of ME-Woodbury.

**Latent Dimension Evaluation** We test the impact of latent dimension $d$ on the performance of Woodbury-Glow. We train our models on CIFAR-10, and use bpd as metric. We vary $d$ within $\{2, 4, 8, 16, 32\}$. The results are in Table 2. When $d < 8$, the model performance will be impacted. When $d > 16$, increasing $d$ will not improve the bpd. This is probably because when $d$ is too small, the latent features cannot represent the input variables well, and when $d$ is too big, the models become hard to train. When $8 \le d \le 16$, the Woodbury transformations are

Table 2: Evaluation of different $d$ (bpd).

|          | Woodbury | ME-Woodbury |
|----------|----------|-------------|
| $d = 2$  | 3.54     | 3.53        |
| $d = 4$  | 3.51     | 3.51        |
| $d = 8$  | 3.48     | 3.48        |
| $d = 16$ | 3.47     | 3.48        |
| $d = 32$ | 3.47     | 3.48        |

powerful enough to model the interactions among dimensions. We also test two values of $d$, i.e., $16, 32$, of Woodbury-Glow on ImageNet $64 \times 64$. The bpds of both $d$ are $3.87$, which are consistent with our conclusion.

**Sample Quality Comparisons** We train Glow and Woodbury-Glow on the CelebA-HQ dataset [19]. We use 5-bit images and set the size of images to be $64\times64$, $128\times128$, and $256 \times 256$. Due to our limited computing resources, we use relatively small models in our experiments. We follow Kingma and Dhariwal [21] and choose a temperature parameter to encourage higher quality samples. Detailed parameter settings are in the appendix. We compare samples from Glow and Woodbury-Glow during three phases of training, displayed in Fig. 4. The samples show a clear trend where Woodbury-Glow more quickly learns to generate reasonable face shapes. After 100,000 iterations, it can already generate reasonable samples, while Glow's samples are heavily distorted. Woodbury-Glow samples are consistently smoother

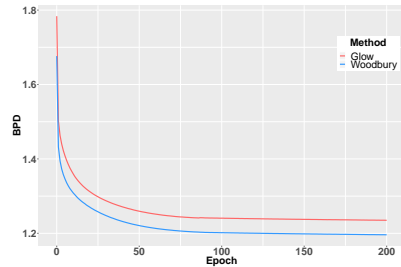

Figure 5: Learning curves on CelebA-HQ 64x64. The NLL of Woodbury Glow decreases faster than Glow.

and more realistic than samples from Glow in all phases of training. The samples demonstrate Woodbury transformations' advantages. The learning curves in Figure 5 also show that the NLL of Woodbury Glow decreases faster, which is consistent to the sample comparisons. In the appendix, we show analogous comparisons using higher resolution versions of CelebA data, which also exhibit the trend of Woodbury-Glow generating more realistic images than Glow at the same training iterations.

## 6   Conclusion

In this paper, we develop Woodbury transformations, which use the Woodbury matrix identity to compute the inverse transformations and Sylvester's determinant identity to compute Jacobian determinants. Our method has the same advantages as invertible $d \times d$ convolutions that can capture correlations among all dimensions. In contrast to the invertible $d \times d$ convolutions, our method is parallelizable and the computational complexity of our methods are linear to the input size, so that it is still efficient in computation when the input is high-dimensional. One potential limitation is that Woodbury transformations do not have parameter sharing scheme as in convolutional layers, so one potential future research is to develop partially Woodbury transformations that can share parameters. We test our models on multiple image datasets and they outperform state-of-the-art methods.

## Broader Impact

This paper presents fundamental research on increasing the expressiveness of deep probabilistic models. Its impact is therefore linked to the various applications of such models. By enriching the class of complex deep models for which we can train with exact likelihood, we may enable a wide variety of applications that can benefit from modeling of uncertainty. However, a potential danger of this research is that deep generative models have been recently applied to synthesize realistic images and text, which can be used for misinformation campaigns.

## Acknowledgments

We thank NVIDIA's GPU Grant Program and Amazon's AWS Cloud Credits for Research program for their support. The work was completed while both authors were affiliated with the Virginia Tech Department of Computer Science. Bert Huang was partially supported by an Amazon Research Award and a grant from the U.S. Department of Transportation Safe-D Program for work on separate projects not directly related to this paper.

## Footnotes

[1] A more general version replaces $\mathbf{I}^{(n)}$ and $\mathbf{I}^{(k)}$ with arbitrary invertible $n \times n$ and $k \times k$ matrices. But this simplified version is sufficient for our tasks.

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
