[Supplementary Material]

# Appendix: Woodbury Transformations for Deep Generative Flows

**You Lu**
Department of Computer Science
Virginia Tech
Blacksburg, VA
you.lu@vt.edu

**Bert Huang**
Department of Computer Science
Tufts University
Medford, MA
bert@cs.tufts.edu

## A  More Background

In this section, we introduce more detailed background knowledge.

### A.1  Normalizing Flows

Let $\mathbf{x}$ be a high-dimensional continuous variable. We suppose that $\mathbf{x}$ is drawn from $p^*(\mathbf{x})$, which is the true data distribution. Given a collected dataset $\mathcal{D} = \{\mathbf{x}_1, \mathbf{x}_2, ..., \mathbf{x}_D\}$, we are interested in approximating $p^*(\mathbf{x})$ with a model $p_\theta(\mathbf{x})$. We optimize $\theta$ by minimizing the negative log-likelihood

$$\mathcal{L}(\mathcal{D}) = \sum_{i=1}^{D} -\log p_\theta(\mathbf{x}_i). \tag{1}$$

For some settings, variable $\tilde{\mathbf{x}}$ is discrete, e.g., image pixel values are often integers. In these cases, we dequantize $\tilde{\mathbf{x}}$ by adding continuous noise $\boldsymbol{\mu}$ to it, resulting in a continuous variable $\mathbf{x} = \tilde{\mathbf{x}} + \boldsymbol{\mu}$. As shown by Ho et al. [3], the log-likelihood of $\tilde{\mathbf{x}}$ is lower-bounded by the log-likelihood of $\mathbf{x}$.

Normalizing flows enable computation of $p_\theta(\mathbf{x})$, even though it is usually intractable for many other model families. A normalizing flow [11] is composed of a series of invertible functions $\mathbf{f} = \mathbf{f}_1 \circ \mathbf{f}_2 \circ ... \circ \mathbf{f}_K$, which transform $\mathbf{x}$ to a latent code $\mathbf{z}$ drawn from a simple distribution. Therefore, with the *change of variables* formula, we can rewrite $\log p_\theta(\mathbf{x})$ to be

$$\log p_\theta(\mathbf{x}) = \log p_Z(\mathbf{z}) + \sum_{i=1}^{K} \log \left| \det \left( \frac{\partial \mathbf{f}_i}{\partial \mathbf{r}_{i-1}} \right) \right|, \tag{2}$$

where $\mathbf{r}_i = \mathbf{f}_i(\mathbf{r}_{i-1})$, $\mathbf{r}_0 = \mathbf{x}$, and $\mathbf{r}_K = \mathbf{z}$.

### A.2  Invertible $d \times d$ Convolutions

Emerging convolutions [4] combine two autoregressive convolutions [2, 8]. Formally,

$$\mathbf{M}_1' = \mathbf{M}_1 \odot \mathbf{A}_1, \qquad \mathbf{M}_2' = \mathbf{M}_2 \odot \mathbf{A}_2, \qquad \mathbf{y} = \mathbf{M}_2' \star (\mathbf{M}_1' \star \mathbf{x}),$$

where $\mathbf{M}_1, \mathbf{M}_2$ are convolutional kernels whose size is $c \times c \times d \times d$, and $\mathbf{A}_1, \mathbf{A}_2$ are binary masks. The symbol $\star$ represents the convolution operator.[1] An emerging convolutional layer has the same receptive fields as standard convolutional layers, which can capture correlations between a target pixel and its neighbor pixels. However, like other autoregressive convolutions, computing the inverse of an

emerging convolution requires sequentially traversing each dimension of input, so its computation is not parallelizable and is a computational bottleneck when the input is high-dimensional.

Periodic convolutions [1, 4] use discrete Fourier transformations to transform both the input and the kernel to Fourier domain. A periodic convolution is computed as

$$\mathbf{y}_{u,:,:} = \sum_v \mathcal{F}^{-1}(\mathcal{F}(\mathbf{M}_{u,v,:,:}^{(p)}) \odot \mathcal{F}(\mathbf{x}_{v,:,:})),$$

where $\mathcal{F}$ is a discrete Fourier transformation, and $\mathbf{M}^{(p)}$ is the convolution kernel whose size is $c \times c \times d \times d$. The computational complexity of periodic convolutions is $\mathcal{O}(c^2 hw \log(hw) + c^3 hw)$. In our experiments, we found that the Fourier transformation requires a large amount of memory. These two problems impact the efficiency of both training and sampling when the input is high-dimensional.

## B   Memory-Efficient Woodbury Transformations

Memory-Efficient Woodbury transformations can effectively reduce the space complexity. The main idea is to perform spatial transformations along the height and width axes separately, i.e., a height transformation and a width transformation. The transformations are:

$$
\begin{aligned}
\mathbf{x}_c &= (\mathbf{I}^{(c)} + \mathbf{U}^{(c)}\mathbf{V}^{(c)})\mathbf{x}, \\
\mathbf{x}_w &= \text{reshape}(\mathbf{x}_c, (ch, w)), \\
\mathbf{x}_w &= \mathbf{x}_c(\mathbf{I}^{(w)} + \mathbf{U}^{(w)}\mathbf{V}^{(w)}), \\
\mathbf{x}_h &= \text{reshape}(\mathbf{x}_w, (cw, h)), \\
\mathbf{y} &= \mathbf{x}_h(\mathbf{I}^{(h)} + \mathbf{U}^{(h)}\mathbf{V}^{(h)}), \\
\mathbf{y} &= \text{reshape}(\mathbf{y}, (c, hw)), \quad\quad\quad (3)
\end{aligned}
$$

where reshape$(\mathbf{x}, (n, m))$ reshapes $\mathbf{x}$ to be an $n \times m$ matrix. Matrices $\mathbf{I}^{(w)}$ and $\mathbf{I}^{(h)}$ are $w$- and $h$-dimensional identity matrices, respectively. Matrices $\mathbf{U}^{(w)}, \mathbf{V}^{(w)}, \mathbf{U}^{(h)}$, and $\mathbf{V}^{(h)}$ are $w \times d_w$, $d_w \times w$, $w \times d_w$, and $d_w \times w$ matrices, respectively, where $d_w$ and $d_h$ are constant latent dimensions.

Using the Woodbury matrix identity and the Sylvester's determinant identity, we can compute the inverse and Jacobian determinant:

$$
\begin{aligned}
\mathbf{y} &= \text{reshape}(\mathbf{y}, (cw, h)), \\
\mathbf{x}_h &= \mathbf{y}(\mathbf{I}^{(h)} - \mathbf{U}^{(h)}(\mathbf{I}^{(d_h)} + \mathbf{V}^{(h)}\mathbf{U}^{(h)})^{-1}\mathbf{V}^{(h)}), \\
\mathbf{x}_w &= \text{reshape}(\mathbf{x}_h, (ch, w)), \\
\mathbf{x}_w &= \mathbf{x}_w(\mathbf{I}^{(w)} - \mathbf{U}^{(w)}(\mathbf{I}^{(d_w)} + \mathbf{V}^{(w)}\mathbf{U}^{(w)})^{-1}\mathbf{V}^{(w)}), \\
\mathbf{x}_c &= \text{reshape}(\mathbf{x}_w, (c, hw)), \\
\mathbf{x} &= (\mathbf{I}^{(c)} - \mathbf{U}^{(c)}(\mathbf{I}^{(d_c)} + \mathbf{V}^{(c)}\mathbf{U}^{(c)})^{-1}\mathbf{V}^{(c)})\mathbf{x}_c, \quad\quad\quad (4)
\end{aligned}
$$

$$
\begin{aligned}
\log\left|\det\left(\frac{\partial \mathbf{y}}{\partial \mathbf{x}}\right)\right| &= hw \log\left|\det(\mathbf{I}^{(d_c)} + \mathbf{V}^{(c)}\mathbf{U}^{(c)})\right| + ch \log\left|\det(\mathbf{I}^{(d_w)} + \mathbf{V}^{(w)}\mathbf{U}^{(w)})\right| \\
&\quad + cw \log\left|\det\left(\mathbf{I}^{(d_h)} + \mathbf{V}^{(h)}\mathbf{U}^{(h)}\right)\right|, \quad\quad\quad (5)
\end{aligned}
$$

where $\mathbf{I}^{(d_w)}$ and $\mathbf{I}^{(d_h)}$ are $d_w$- and $d_h$-dimensional identity matrices, respectively. The Jacobian of the reshape() is an identity matrix, so its log-determinant is 0.

We call Equation 3 the memory-efficient Woodbury transformation because it reduces space complexity from $\mathcal{O}(c + hw)$ to $\mathcal{O}(c + h + w)$. This method is effective when $h$ and $w$ are large. To analyze its complexity, we let all latent dimensions be less than $d$ as before. The complexity of forward transformation is $\mathcal{O}(dchw)$; the complexity of computing the determinant is $\mathcal{O}(d(c + h + w) + d^3)$; and the complexity of computing the inverse is $\mathcal{O}(dchw + d^2(c + ch + cw) + d^3)$. The same as Woodbury transformations, when the input is high dimensional, we can omit $d$. Therefore, the computational complexities of the memory-efficient Woodbury transformation are also linear with the input size.

We list the complexities of different methods in Table 1. We can see that the computational complexities of Woodbury transformations are comparable to other methods, and maybe smaller when the input is high-dimensional, i.e., the $c, h, w$ are big.

Table 1: Comparisons of computational complexities.

| Method | Forward | Backward |
|---|---|---|
| 1x1 convolution | $\mathcal{O}(c^2hw + c^3)$ | $\mathcal{O}(c^2hw)$ |
| Periodic conolution | $\mathcal{O}(chw\log(hw) + c^3hw)$ | $\mathcal{O}(chw\log(hw) + c^2hw)$ |
| Emerging convolution | $\mathcal{O}(c^2hw)$ | $\mathcal{O}(c^2hw)$ |
| ME-Woodbury transformation | $\mathcal{O}(dchw)$ | $\mathcal{O}(dchw)$ |
| Woodbury transformation | $\mathcal{O}(dchw)$ | $\mathcal{O}(dchw)$ |

## C  Parameter Settings

In this section, we present additional details about our experiments to aid reproducibility.

### C.1  Experiments of Quantitative Evaluation

In the experiments of qualitative evalution, we compare Woodbury transformations with 3 permutation layer baselines, i.e., 1x1 convolution, emerging convolution, and periodic coupling, and 2 coupling layer baselines, i.e., neural spline coupling, and MaCow. For all generalized permutation methods, we use affine coupling, which is composed of 3 convolutional layers, and the 2 latent layers have $512$ channels. For the neural spline coupling, we set the number of spline bins to $4$. The spline parameters are generated by a neural network, which is also composed of convolutional layers. For $32 \times 32$ images, we set the number of channels to $256$, and for $64 \times 64$ images, we set it to $224$. Ma et al. [10] used steps containing a MaCow unit, i.e., 4 autoregressive convolution coupling layers, and a full Glow step. For fair comparison, we directly use the MaCow unit to replace the affine coupling. For $32 \times 32$ images, we set the convolution channel to $384$, and for $64 \times 64$ images, we set it to $296$.

We run each method to fixed number of iterations and test it every $10,000$ iterations. The bpds reported in our main paper are the best bpds obtained by each method. The bpds are single-run results. This is because each run of the experiment requires 3 to 5 days, and running each model multiple times is a major cost. We found in our experiments that for the same model and parameter settings, the bpds' standard deviation of multiple runs are very small, i.e., around $0.003$, so single run results are sufficient for comparing bpd.

### C.2  Hyper-parameter Settings

We use Adam [6] to tune the learning rates, with $\alpha = 0.001$, $\beta_1 = 0.9$, and $\beta_2 = 0.999$. We use uniform dequantization. The sizes of models we use, and mini-batch sizes for training in our experiments are listed in Table 2.

Table 2: Model sizes and mini-batch sizes.

| Dataset | Mini-batch size | Levels(L) | Steps(K) | Coupling channels |
|---|---|---|---|---|
| CIFAR-10 32x32 | 64 | 3 | 8 | 512 |
| ImageNet 32x32 | 64 | 3 | 8 | 512 |
| ImageNet 64x64 | 32 | 4 | 16 | 512 |
| LSUN Church 96x96 | 16 | 5 | 16 | 256 |
| CelebA-HQ 64x64 | 8 | 4 | 16 | 512 |
| CelebA-HQ 128x128 | 4 | 5 | 24 | 256 |
| CelebA-HQ 256x256 | 4 | 6 | 16 | 256 |

### C.3  Latent Dimension Settings

In all our experiments, we set the latent dimensions of Woodbury transformations, and ME-Woodbury transformations as in Table 3.

Table 3: Latent dimensions of Woodbury transformations and ME-Woodbury transformations. The numbers in the brackets represent the latent dimension used in that level. For example, the $d_c$ : $\{8, 8, 16\}$, represents that the settings of $d_c$ at the three levels are $8$, $8$, and $16$.

| Dataset | Woodbury | ME-Woodbury |
|---|---|---|
| CIFAR-10 32x32 | $d_c : \{8, 8, 16\}$ <br> $d_s : \{16, 16, 8\}$ | $d_c : \{8, 8, 16\}$ <br> $d_h : \{16, 16, 8\}$ <br> $d_w : \{16, 16, 8\}$ |
| ImageNet 32x32 | $d_c : \{8, 8, 16\}$ <br> $d_s : \{16, 16, 8\}$ | $d_c : \{8, 8, 16\}$ <br> $d_h : \{16, 16, 8\}$ <br> $d_w : \{16, 16, 8\}$ |
| ImageNet 64x64 | $d_c : \{8, 8, 16, 16\}$ <br> $d_s : \{16, 16, 8, 8\}$ | $d_c : \{8, 8, 16, 16\}$ <br> $d_h : \{16, 16, 8, 8\}$ <br> $d_w : \{16, 16, 8, 8\}$ |
| LSUN Church 96x96 | $d_c : \{8, 8, 16, 16, 16\}$ <br> $d_s : \{16, 16, 16, 8, 8\}$ | — |
| CelebA-HQ 64x64 | $d_c : \{8, 8, 16, 16\}$ <br> $d_s : \{16, 16, 8, 8\}$ | — |
| CelebA-HQ 128x128 | $d_c : \{8, 8, 16, 16, 16\}$ <br> $d_s : \{16, 16, 16, 8, 8\}$ | — |
| CelebA-HQ 256x256 | $d_c : \{8, 8, 16, 16, 16, 16\}$ <br> $d_s : \{16, 16, 16, 16, 8, 8\}$ | — |

## D   Sample Quality Comparisons

We compare the samples generated by Woodbury-Glow and Glow models trained on the CelebA-HQ dataset. We follow Kingma and Dhariwal [7] and randomly hold out 3,000 images as a test set. We use 5-bits images. We use $64 \times 64$, $128 \times 128$, $256 \times 256$ images. Due to the our limited computing resources, we use relatively small models. The model sizes and other settings are listed in Table 2 and Table 3. We generate samples from the models during different phases of training and display them in Figure 1, and Figure 2 (The results of $64 \times 64$ images are shown in the main paper). For the $128 \times 128$ images, both Glow and Woodbury-Glow generate distorted images at iteration 100,000, but Woodbury-Glow seems to improve in later stages, stabilizing the shapes of faces and structure of facial features. Glow, continues generating faces with distorted overall shapes as training continues. For the $256 \times 256$ images, neither model ever trains sufficiently to generate highly realistic faces, but Woodbury-Glow makes significantly more progress in these 300,000 iterations than Glow. Glow's samples at 300,000 are still mostly random swirls with an occasional recognizable face, while almost all of Woodbury-Glow's samples look like faces, though distorted. Due to limits on our computational resources, we stopped the higher resolution experiments at 300,000 iterations (rather than running to 600,000 iterations as we did for the $64 \times 64$ experiments in the main paper). With a larger model and longer training time, it seems Woodbury-Glow would reach higher sample quality much faster than Glow.

The likelihoods of test set under the trained model are listed in Table 3. For the $64 \times 64$ and $128 \times 128$ images, Woodbury-Glow scores higher likelihood than Glow. For the $256 \times 256$ images, their likelihoods are almost identical, and are better than the score reported in [7]. This may be due to three possible reasons: (1) We use affine coupling rather than additive coupling, which is a non-volume preserving layer and may improve the likelihoods; (2) Since the test set is randomly collected, it is different from the one used in [7]; And (3) The model used in [7] is very large, so it may be somewhat over-fitting. Surprisingly, the clear difference in sample quality is not reflected by the likelihoods. This discrepancy may be because we use 5-bit images, and the images are all faces, so the dataset is less complicated than other datasets such as ImageNet. Moreover, even though Glow cannot generate reasonable $256 \times 256$ samples, the colors of these samples already match the colors of real images well, so these strange samples may non-intuitively be equivalently likely as the face-like samples from Woodbury-Glow.

Table 4: Bit per-dimension results on CelebA-HQ

| Size of images | Glow | Woodbury-Glow |
|---|---|---|
| $64 \times 64$ | 1.27 | **1.23** |
| $128 \times 128$ | 1.09 | **1.04** |
| $256 \times 256$ | **0.93** | **0.93** |

Figure 1: Random samples of $128 \times 128$ images drawn with temperature $0.7$ from a model trained on CelebA data.

Figure 2: Random samples of $256 \times 256$ images drawn with temperature $0.7$ from a model trained on CelebA data.

# E   Additional Samples

In this section, we include additional samples from Woodbury-Glow models trained on our various datasets. These samples complement our quantitative analysis. We train our models on CIFAR-10 [9], ImageNet [12], the LSUN church dataset [13], and the CelebA-HQ dataset [5]. Specifically, for ImageNet, we use $32 \times 32$ and $64 \times 64$ images. For the LSUN dataset, we use the same approach as Kingma and Dhariwal [7] to resize the images to be $96 \times 96$. For the CelebA-HQ dataset, we use $64 \times 64$, $128 \times 128$, and $256 \times 256$ images. For LSUN and CelebA-HQ datasets, we use 5-bit images. The parameter settings of our models are in Table 2 and Table 3. The samples are in Figures 3, 4, 5, 6, 7, 8, and 9.

Figure 3: CIFAR-10 $32 \times 32$ Woodbury-Glow samples.

Figure 4: ImageNet $32 \times 32$ Woodbury-Glow samples.

Figure 5: ImageNet $64 \times 64$ Woodbury-Glow samples.

Figure 6: LSUN church $96 \times 96$ Woodbury-Glow samples (temperature $0.875$).

Figure 7: CelebA-HQ $64 \times 64$ Woodbury-Glow samples (temperature $0.7$).

Figure 8: CelebA-HQ $128 \times 128$ Woodbury-Glow samples (temperature $0.5$).

Figure 9: Selected CelebA-HQ $256 \times 256$ Woodbury-Glow samples (temperature $0.5$).

## Footnotes

[1] In practice, a convolutional layer is usually implemented as an aggregation of cross-correlations. We follow Hoogeboom et al. [4] and omit this detail.