[Reviews · NeurIPS 2020]

Review 1

Summary and Contributions: This paper utilises Woodbury's matrix identity and Sylvester's determinant identity to construct low-rank linear flows on high dimensional spaces, which they call Woodbury Transformations. The paper empirically compares running-time and NLL modelling on CIFAR10, Imagenet32/64 and CelebA in a glow-type generative model by comparing their Woodbury transforms with 1x1 convolutions and other alternatives in literature.

Strengths: The paper provides a simple but effective method to construct tractable linear flows on high-dimensional spaces. The experimental section describes trade-offs in running-time of different methods in literature and show improved NLL performance when their method replaces 1x1 convolutions or other methods. Even though other methods obtain similar NLL performance, the running-time experiment shows the advantage of utilising Woodbury transformation in these situations.

Weaknesses: The method is relatively straightforward and some directions could be explored further. For instance, by using fully connected transformations, typical convolutional weight-sharing is not utilised. It would be good to discuss downsides to fully connected Woodbury transforms, and possible alternative formulations that would utilise convolutional weight-sharing. Further, the models utilised in the experimental section are quite small. As a result, the NLL performance is not very good compared to newer flow-based models. In addition the gains in NLL are quite small, and it would be better if the authors included standard deviations over multiple runs. Minor: - If possible, I would advice the authors to include the "changing bottleneck" experiment in the main paper. This experiment relates to the required size of the bottleneck. - For a better overview it would be nice to have a table showing the complexity for different methods for their forward/inverse/logdet in one place.

Correctness: The paper builds upon two existing identities from literature and discusses time-complexity for their method. These seem to be correct. On the empirical evaluation the paper could be more precise: Please describe in detail how the best performing model was selected (i.e. run for fixed epochs, or select best based on validation set). Are the results shown in Table 1 single-run? Or an average over multiple runs?

Clarity: Overall, the paper is well-written. The comments on time-complexity throughout the paper are appreciated.

Relation to Prior Work: In general the relation to prior work is clear. Note that in [1] a somewhat similar approach was used to obtain rank-1 covariance matrices for variational auto encoders. Perhaps this would be a nice addition to the related work. Also, the above-mentioned disadvantage to convolutional-weight sharing could be discussed with respect to emerging and periodic convolutions. [1] Stochastic Backpropagation and Approximate Inference in Deep Generative Models. Danilo J. Rezende, Shakir Mohamed, Daan Wierstra.

Reproducibility: Yes

Additional Feedback: ==== After rebuttal ===== I am mostly happy with the rebuttal and I have updated my score to a 7. I strongly advice the authors to take into account the following recommendations and update their paper accordingly: - The authors argue in the rebuttal that Squeeze layers also destroy spatial weight sharing and therefore it doesn’t matter that Woodbury transforms do not have spatial weight sharing. Although I agree that squeeze layers do this (only to some degree though), I would really like to see the authors mentioning this as a possible limitation to the Woodbury transforms. - From the description in the rebuttal, it seems that the authors report the best performing model by looking at the test performance every T iterations. Although overfitting in generative modelling is by far not as bad as in a task like classification, reporting test performance in this manner is not good practice. Perhaps it would be good to highlight in the paper if this is indeed the case, and note that this is not the best approach to model selection.


Review 2

Summary and Contributions: The paper discusses a new class of invertible transformations for flow-based generative model. The idea his to utilize Woodbury transformations. The authors propose to utilize channel and spatial transformations to achieve flexible and efficient transformations. Further, they show how the Woodbury matrix identity allows to invert channel/spatial transformation, and Sylvester's determinant identity to obtain the Jacobian determinant. The empirical results show potential of the presented idea.

Strengths: + The proposed spatial and channel transformations, and their parameterization to allow relatively easy inversion is interesting. + The manner the matrices are expressed allows to utilize the Woodbury identity, and, thus, the transformation is invertible. + The empirical evaluation is sound. + The proposed transformations could be useful as a building block of invertible neural networks and flow-based models. + The memory efficient version of the proposed transformations allows to reduce the number of parameters while maintaining almost the same performance.

Weaknesses: - I wonder what is the total computational complexity compared to other methods (e.g., emerging convolutions). If I imagine the Woodbury flow working on a mobile device, the number of operations could cause a significant power demand. - Following on that, I am worried that the total computational complexity is much higher for other approaches. This could limit the usability of the proposed transformation.

Correctness: I do not find any flaws of the presented methods. Similarly, I find the experiments sound and well performed. The paper is very well written. The organization is correct, the flow is good. All concepts are clearly explained and easy to follow.

Clarity: The paper is very well written. The organization is correct, the flow is good. All concepts are clearly explained and easy to follow.

Relation to Prior Work: The paper explains precisely what is the prior work.

Reproducibility: Yes

Additional Feedback: * The only comment I can think of is about the total computational complexity. Maybe it would be fair to add it in the appendix. ===AFTER THE REBUTTAL=== I would like to thank the authors for their rebuttal. In my opinion the paper is good and solid, and it deserves to be accepted. I was leaning towards 8 even, however, after a vidid discussion with other reviewers, I decided to keep my score.


Review 3

Summary and Contributions: The authors propose to use the Woodbury matrix identity and Sylvester’s determinant identity to effectivly compute the inverse and Jacobian determinant in the deep generative flows models. One of the benefits of using these mechanisms is to accelerate model learning. Moreover, a Woodbury transformation can find deeper dependencies of the features because it are able to model correlations along both channel and spatial axes.

Strengths: This paper makes an empirical contribution.

Weaknesses: The presented empirical results do not confirm the significant advantage of the introduced modifications in relation to the already existing methods. Detailed comments to the results: 1) In Fig. 4 the results of Glow method do not coincide with the results that were presented in the original paper. I supposed that both models were trained too short. 2) Fig. 3 shows that almost in all cases 1x1 convolutions wins. Based on these results, one can say that the application of Woodbury transformations in flow models does not significantly reduce training time. Moreover, the authors state that "NLL of Woodbury Glow decreases faster". I would say that it obtains lower values but the decrease is similar, Fig 5. In my opinion, presented results are insufficient for the NeurIPS.

Correctness: I have discussed most of my concerns above.

Clarity: Yes

Relation to Prior Work: Yes

Reproducibility: Yes

Additional Feedback: Post rebuttal ============================================== Thanks for the author's response. Following the comments of the other reviewers. I agree that the paper is clearly written, the idea is sound. Hence I changed my score to 6. I didn't give a higher rating because the authors took the best performing model (according to them) and not compare with the model with full capacity as proposed in the original papers.


Review 4

Summary and Contributions: The paper introduces an efficient transformation. This model constructs a dense fully connected layer with a low rank plus identity matrix which enjoys the efficient inversion and determinant computation of the Woodbury matrix identity and Sylvester’s determinant identity. A memory efficient version of the transformation was also introduced which factorizes the filtering along the space into two independent so called Woodbury transformations along each axis.

Strengths: The paper introduces an efficient transformation to the library of invertible flows. Empirical evaluations show that the proposed model can slightly improve the bpd performance. Comparing the proposed transformation with the invertible convolutions with complete kernel size of n, eg those used in [17], the proposed Woodbury transformation has 2*n*d parameters so it can offer more flexibility in general but on the other hand it is not specifically designed for locally structured inputs such as images as the convolutions are.

Weaknesses: After reading the rebuttal, I have updated my score but I believe it is necessary that paper clarify that this model is a simple form of Sylvester flow and compare it with a sequence of planar flow as, roughly speaking, the Woodburry transformation can also be viewed as applying the planar flow sequentially (d times). i.e. y = (I + U V^T)x = (I + u_1 * v_1^T + u_2 * v_2^T + ... + u_d * v_d^T) where v_i is i^{th} column of matrix V. The proposed transformation is indeed a low-rank plus identity transformation which can be seen as a simple form of Sylvester Flow, z’ = z + Ah(Bz + b) where b=0 and h(x) = x. The description given for Sylvester Flow in section 4 is inaccurate. It can actually be performed on matrix (tensor) inputs and also can be performed along each axis in the same way as ME-Woodbury. Its inverse will be tractable and there has analytical form for inversion by making some simplifications, eg reducing it to Woodbury flow, otherwise the inversion is possible using iterative methods, so it *can* generate samples efficiently. So my main concern is the limited novelty of the proposed model in comparison to the available flows but I may be persuaded during the rebuttal phase.

Correctness: Throughout the paper, the invertible convolutions are claimed to be computationally inefficient and the computational cost of the circular (periodic) convolution is mentioned inaccurately. To be more accurate, for 2-D inputs of size N1*N2, the Fourier transform can be computed in O(N1*N2(logN1 + logN2)) time. For more detail please refer to [17] or Rafael C Gonzalez and Richard E Woods. Digital image processing, 1992. Interestingly, computational complexity of Woodbury flow for input of size 256*256 with d=16 is of the same order as that of 2-D convolution since (logN1 + logN2) =16. Woodbury flow can offer more flexibility in some datasets but it does not take into account local structure that a convolution can capture in 2-D images. The running time experiments in section 5 do not sound a fair comparison between different methods as the models dont have similar numbers of parameters. For example emerging convolution of 3*3 is not comparable with Woodbury of n*d parameters. Also, it is expected that ME-Woodbury be faster than the Woodbury as it replaced a heavier matrix products of size h*w*d with a two lighter products along each of sizes h*d+w*d. Furthermore, as mentioned above, the periodic (circular) convolution is also computationally efficient and their running time performance depends on employing a fast and parallelizable implementation of 2-D FFT. The results in Table 1, show that the proposed flows can slightly enhance the performance compared to baselines. But the shortcoming is that small models are trained for all the models rather than the model with full capacity as proposed in the original paper and hence the results are very far from those in the original paper. So to support your conclusion -- the Woodbury will make new SOTA results -- it is also important to reproduce the models with their full capacity and make a comparison with your model with similar complexity.

Clarity: The presentation is good but can be improved. Here are some details and possible typos: Vertical Space is needed after the caption of Figure 1 to separate it from the text. Lines 172-176 are repeated twice. Line 180: the correct Sylvester flow has an extra identity mapping (scape connection), ie z_{t+1}= z_{t} + … . Line 185: the statement “Sylvester flows are inverse functions” is not clear.

Relation to Prior Work: It provided a good review of available NFs but it also needs to re-examine the main advantage of the proposed model over Sylvester flow.

Reproducibility: Yes

Additional Feedback:

[Author Response · NeurIPS 2020]

We thank the reviewers for their insightful comments. Due to limited space, we respond to your main points below.

**[Reviewer 1]** 1. The goal of our experiments is to test that, under the same settings, models with Woodbury layers outperform other baselines—not to aim for SOTA bpds. We use smaller models to budget computational resources to test the key hypotheses. As R2 wrote, our experiments fairly compare all models with the same settings. In terms of the bpd improvements, we argue that they are significant enough to demonstrate the improvements of our method. Emerging convolution Hoogeboom et al. [2019] (Table 4) improves 0.02 bpd over 1x1 convolution on ImageNet32. Our experiments show an even more significant improvement: 0.12 bpd on ImageNet32. Finally, as R2 suggested, evaluation with small models is informative and important for low-resource settings, such as mobile devices.

2. We will add expanded experimental details to the appendix. We run each model to fixed number of iterations: 300,000 iterations and around 380 epochs for CIFAR10 and 450,000 iterations and around 20 epochs for ImageNet. We test each method every fixed number of iterations. The bpds in Table 1 are the best bpds obtained by each method. The bpds are single-run results, since each run of the experiment requires 3 to 5 days, and we test 7 models on 3 datasets. So running each model multiple times is a major cost. We tune the hyper-parameters for each model, e.g., weight decay of the optimizer and gradient norm, with a grid search and only running 10,000 iterations, before we run the full training.

3. We will follow your suggestions to move the latent dimension bottleneck experiment to the main paper and list the complexities of methods in a table. Thanks for your ideas for further exploration. We'll add these to our discussion.

**[Reviewer 2]** 1. We appreciate your positive comments. Let the input be a $c \times h \times w$ tensor. The complexities are $\mathcal{O}(c^2 hw + c^3)$ for 1x1 convolutions, $\mathcal{O}(c^2 hw)$ for emerging convolutions, and $\mathcal{O}(chw \log(hw) + c^3 hw)$ for periodic convolutions. The Woodbury layer complexity is $\mathcal{O}(dchw)$, where $d$ is the latent dimension and can be roughly seen as constant. The complexity of Woodbury layers is comparable to other methods and even smaller than periodic convolutions when $c, h, w$ are big. Therefore, it should be possible to apply Woodbury layers on mobile devices.

**[Reviewer 3]** 1. We believe that R3 has misunderstood some important points. Woodbury transformations are not meant to be faster than 1x1 convolutions. Instead, they are a low-cost way to model richer interactions among dimensions. Figure 3 shows that Woodbury layers are more efficient than other richer layers in either training or sampling. Figure 5 shows that Woodbury models score better bpd. We will clarify our text to help other readers avoid this confusion.

2. To obtain thorough comparisons with reasonable computational budget, we trained smaller models than those in the original Glow paper. We trained the models long enough to fairly compare the different methods. Kingma and Dhariwal use a very large model and train on 40 GPUs. Our results in Figure 4 use 64x64 images, and Kingma and Dhariwal use 256x256 images. We train each model for 200 epochs, and the curves in Figure 5 suggest that the training converged.

**[Reviewer 4]** 1. We will be more careful about describing Sylvester flows and explaining how Woodbury flows differ. We did not mean to claim that Sylvester flows cannot be applied to tensors. We meant to say that Berg et al. only (theoretically and empirically) analyze Sylvester's flows on vectors. There is no published application of Sylvester's flows on high-dimensional tensors. So one novelty of Woodbury flows is that they are designed for high-dimensional tensors, with channel transformations and spatial transformations, and two ways, i.e., Woodbury and ME-Woodbury, to combine them. We agree that $z' = z + Ah(Bz + b)$ defines a very general set of flows, but one needs to define efficient methods to use specific instantiations of it. Berg et al. analyze one special variant of it, with $A = QR$, and $B = \tilde{R}Q^T$. Woodbury flows is another variant that allows tractable inversion. These are the key novelties of Woodbury flows when compared to the flows analyzed in Berg et al..

2. Thank you for pointing out another way to compute FFT. We report the FFT complexity for the method described by Hoogeboom et al. [2019] (footnote 2). In our code, we directly use the PyTorch FFT implementation. We agree a better FFT algorithm can theoretically make the periodic convolutions faster, and we already mentioned it in the paper (line 239). We will make this caveat clearer when discussing the comparision to periodic convolution.

3. About the running time comparisons, these are measurements of running time for models used in the other experiments to provide a view of the relationship between running time and modeling power. The methods all scale differently with parameter size, but we did not empirically test that because it is easily analyzed theoretically based on the complexities of each method's matrix operations. It's more important to compare the cost of methods that are tuned to best model the data. The ME variant only improves memory storage, and it does not save running time because it still has a bottleneck of a matrix product the same size as the full Woodbury. (See Appendix B for details.)

4. About the local structure, the squeeze layers in flows cause true convolution operations to no longer have the same spatial interpretation as in traditional CNNs. So no current flow architectures have nice spatial interpretation.

5. We did not claim to achieve or target SOTA bpds. We compare with modern flow layers under the same settings, and Woodbury transformations outperform them. More discussions are in our response to Reviewer 1.

[Meta-Review · NeurIPS 2020]

The paper proposes to parameterize a linear transformation as a low-rank update to an identity matrix, and then use the Woodbury matrix identity to efficiently compute its inverse and the Sylvester determinant identity to efficiently compute its determinant. Some reviewers expressed concerns regarding novelty, which I share. Indeed, the proposed linear flows are a fairly straightforward application of well-known matrix-algebra techniques for inverting and calculating the determinant of low-rank updates. In fact, TensorFlow probability already uses the Woodbury matrix identity and the Sylvester determinant identity in their `Affine` bijector: https://www.tensorflow.org/probability/api_docs/python/tfp/bijectors/Affine For that reason, I doubt this paper contains much new information for normalizing-flow experts, although it may be useful to a broader machine-learning audience. Having said that, this paper is well-written and well-executed, contains some novel extensions to the basic idea, and it's likely the first published version of Woodbury flows with careful experimental comparisons to alternatives. For that reason, and given that the reviews are mostly positive, I've decided to recommend acceptance of the paper. Nonetheless, the reviewers have raised a few concerns that I would encourage the authors to take to heart when preparing the camera-ready version. In particular, given that the differences between methods are very small, error bars are necessary to judge significance. Also, even though we are not used to citing software, I would encourage the authors to at least mention that similar ideas have been used by TensorFlow probability before.